# Extracellular Vesicles Inhibit the Response of Pancreatic Ductal Adenocarcinoma Cells to Gemcitabine and TRAIL Treatment

**DOI:** 10.3390/ijms23147810

**Published:** 2022-07-15

**Authors:** Ella Rimmer, Sadaf Rashid, Igor Kraev, Francesc Miralles, Androulla Elia

**Affiliations:** 1Pancreatic Cancer Research Group, St George’s University of London, Cranmer Terrace, London SW17 ORE, UK; erimmer@sgul.ac.uk (E.R.); sadaf_1995@hotmail.co.uk (S.R.); fmiralle@sgul.ac.uk (F.M.); 2Electron Microscopy Suite, Faculty of Science, Technology, Engineering and Mathematics, Open University, Walton Hall, Milton Keynes MK7 6AA, UK; igor.kraev@open.ac.uk

**Keywords:** pancreatic cancer, chemoresistance, extracellular vesicles, gemcitabine, apoptosis

## Abstract

Pancreatic ductal adenocarcinoma remains an aggressive cancer with a low 5-year survival rate. Although gemcitabine has been a standard treatment for advanced pancreatic cancer, patients often develop resistance to this therapeutic. We have previously shown that treating pancreatic cancer cells in vitro with a combination of gemcitabine and the cytokine TRAIL significantly reduced both cell viability and survival. The data presented here demonstrate that this response to treatment is inhibited when cells are incubated with a conditioned medium derived from untreated cells. We show that this inhibition is specifically mediated by extracellular vesicles present in the conditioned medium, as seen by a significant decrease in apoptosis. Additionally, we further demonstrate that this effect can be reversed in the presence of GW4869, an inhibitor of exosome biogenesis and release. These results show that pancreatic cancer cell-derived extracellular vesicles can confer resistance to treatment with gemcitabine and TRAIL. The implications of these findings suggest that removal of EVs during treatment can improve the response of cells to gemcitabine and TRAIL treatment in vitro.

## 1. Introduction

Pancreatic ductal adenocarcinoma (PDAC) is often detected at a late stage. The nucleoside analogue gemcitabine has, until recently, been a first-line treatment for pancreatic cancer (PC) for several years [1], and patients have been shown to have an improved quality of life following therapy [2]. Despite this, gemcitabine only elicits minor therapeutic responses, and the overall 5-year survival rate remains around 9% [3]. Several preclinical studies have indicated that using gemcitabine in combination with other therapeutics can improve the response of tumour masses to treatment [4,5], reviewed by Miller et al. [6]. One such treatment is the pro-apoptotic cytokine known as tumour necrosis factor-related apoptosis-inducing ligand (TRAIL), which specifically targets cancer cells [7]. We have previously investigated the sensitivity of PDAC cells to TRAIL-induced apoptosis and found that when combined with gemcitabine, TRAIL enhances the inhibition of survival of those cells in vitro [8]. Although the tumour microenvironment is an important hallmark, containing a plethora of immune cells [9], the absence of the latter when using cells in vitro must be an important consideration. We were therefore interested in investigating the potential role of the conditioned media (CM) taken from untreated cells in the response of PDAC cells to treatment. To carry this out, we initially examined the effect of media conditioned by cell lines with differing responses to gemcitabine and TRAIL treatment. CM broadly contains two fractions: the soluble fraction, comprising cytokines and nutrients, and an insoluble fraction, comprising extracellular vesicles. Some work has been carried out considering the effects of PDAC-derived CM on other cell types, in terms of proteomic changes in the recipient cells [10] and cellular differentiation [11]. Little to no work has been completed to investigate the role of PDAC-cell-derived CM in PDAC cells in vitro. 

We have also specifically investigated the role of the insoluble fraction of CM, namely the extracellular vesicles (EVs). EVs are any lipid-bound vesicles released by cells that can be defined, mainly by their size and contents, into one of the following categories: micro-vesicles, apoptotic bodies and small EVs, such as exosomes [12]. The latter have a diameter of approximately 50–150 nm [12] and carry a range of cargo, including protein, RNA, miRNA and DNA [13]. While the function of these vesicles is yet to be fully elucidated, tumour-derived exosomes have been heavily implicated in the progression and response to treatment of a number of different cancers, including PDAC [14,15,16].

In this study, we investigated the comparative effects of CM and EVs on response of PC cells to a combination treatment with gemcitabine and TRAIL. We found that CM and EVs are capable of reducing the response of recipient cells to treatment with gemcitabine and TRAIL. We also present evidence that, more specifically, when exosome biogenesis and release are inhibited using GW4869, this resistance is reversed. The findings have an important implication for the response of PDAC to gemcitabine and TRAIL treatment in vitro, highlighting the potential role of the tumour microenvironment, even in 2D single-cell-type cultures. 

## 2. Results

### 2.1. TRAIL Enhances the Cytotoxic Effects of Gemcitabine Treatment on Human PC Cells

We have previously investigated the effects of the chemotherapeutic agent gemcitabine in combination with TRAIL on PDAC cell survival [8]. Here, we examined the effects of this combination following the exposure of gemcitabine-treated cells to 100 ng/mL TRAIL over increasing treatment times. Using the thiazolyl blue tetrazolium bromide (MTT) assay, we characterised cell survival in three PDAC cell lines: BxPC-3, MIA PaCa-2 and PANC-1, following treatment with a combination of gemcitabine and TRAIL, as shown in Figure 1a–c respectively. TRAIL (100 ng/mL) was added at several intervals for the final 6 h of the 24 h treatment with 100 µM gemcitabine. All three cell lines demonstrated a variable response over a 6 h period of treatment with the cytokine. Both the BxPC-3 and MIA PaCa-2 cell lines showed a significant decrease in the percentage of cell survival after 4 h of TRAIL treatment: MIA PaCa-2 was the most sensitive of the cell lines, exhibiting 41.4% cell survival at 4 h (Figure 1b) compared with BxPC-3, which had 48.9% survival after 4 h of TRAIL treatment. PANC-1 cells were far more resistant, with only a significant decrease in cell survival observed at 6 h of TRAIL treatment, equating to 86.4% cell survival (Figure 1c). 

### 2.2. PDAC Cell-Conditioned Medium Alters the Response of Other PDAC Cells Lines to Treatment with Gemcitabine and TRAIL

We were interested in examining the effect of PDAC cell-derived CM on PDAC cells’ response to the gemcitabine and TRAIL combination treatment. As the latter is a well-characterised inducer of apoptosis, we used the trypan blue exclusion assay to assess the effect of co-treatment on cell viability. We focused our studies on two of the cell lines that showed a differing response to gemcitabine and TRAIL treatment, as shown in Figure 1a–c. As the PANC-1 cell line was less responsive to TRAIL, we decided to compare the response of this cell line with that of the TRAIL-sensitive cell line MIA PaCa-2. We treated both cell lines with 100 µM gemcitabine for 24 h and 100 ng/mL TRAIL for the final 4 h in MIA PaCa-2 or 6 h in PANC-1 cell lines. The effect of this treatment on cell viability was determined in the presence or absence of PDAC-cell-derived CM from the different cell lines. Figure 2a,b show the effect on the cellular response to treatment with gemcitabine and TRAIL in the presence of CM in the MIA PaCa-2 and PANC-1 cell lines. For example, CM derived from untreated MIA PaCa-2 cells was used to incubate PANC-1 cells during treatment with gemcitabine and TRAIL. Figure 2a demonstrates that, in the presence of the MIA PaCa-2-derived CM, there was a small but non-significant decrease in the response, as denoted by an increase in cell viability following treatment with gemcitabine and TRAIL (58.7% cell viability versus 62.9% cell viability with CM). The effects of MIA PaCa-2-derived CM on the morphology of PANC-1 cells during treatment with and without gemcitabine and TRAIL are shown in Figure 2c. Here, we can see that incubating PANC-1 cells with MIA PaCa-2-derived CM alone in the absence of any treatment appeared to increase the number of rounded cells, although the trypan blue assay shown in Figure 2a suggested that this effect was not statistically significant. Similarly, there was also an increase in the number of rounded PANC-1 cells when treatment with gemcitabine and TRAIL was carried out in the presence of MIA PaCa-2. 

CM derived from untreated PANC-1 cells was included during the treatment of MIA PaCa-2 cells with 100 µM gemcitabine and 100 ng/mL TRAIL. The effect of this CM on cell viability is shown in Figure 2b,d. Here, it can be seen that in the presence of PANC-1-derived CM, MIA PaCa-2 cells were more resistant to treatment with gemcitabine and TRAIL, as denoted by the significant increase in the cell viability (72.2% cell viability versus 60.0% cell viability without CM; Figure 2b). Concordantly, there was also a significant decrease in the percentage of MIA PaCa-2 cells undergoing apoptosis under these conditions, as determined using time-lapse microscopy, (80.8% of cells versus 95.8% of cells without PANC-1 CM; Figure 2d). 

### 2.3. PDAC-Cell-Derived EVs Confer Resistance to Treatment with Gemcitabine and TRAIL

To further investigate the effects of CM on cell viability, we decided to purify the EV fraction of PDAC-cell-derived CM. These were isolated from conditioned media taken from PANC-1 and MIA PaCa-2 cells using differential ultracentrifugation, for use during treatment with 100 µM gemcitabine and 100 ng/mL TRAIL. EVs derived from PANC-1 cells were added to gemcitabine-treated MIA PaCa-2 cells during treatment with TRAIL. This addition decreased the response to the combination treatment, as denoted by the increased viability from 60.0% to 76.2%, as determined using the trypan blue exclusion assay (Figure 3a). This trend was also observed when using MTT assays to establish cell survival (Figure 3b). Here, there was an increase in the survival of MIA PaCa-2 cells following treatment with gemcitabine and TRAIL from 80.0% to 88.6% in the presence of PANC-1-derived EVs. Additionally, using whole-cell extracts from these treatments, we were able to establish a decrease in the cleavage of caspase-8 (Figure 3c), indicating a reduction in apoptosis when PANC-1-derived EVs were added to the TRAIL. Concordantly, MIA PaCa-2-derived EVs were added to gemcitabine-treated PANC-1 cells during the TRAIL treatment. Figure 3d demonstrates that PANC-1 cells were significantly more viable following treatment with gemcitabine and TRAIL in the presence of MIA PACa-2-derived EVs (65.6% viable versus only 52.1% without EVs). This was corroborated with time-lapse microscopy, where the percentage of apoptosis following gemcitabine and TRAIL treatment of PANC cells significantly decreased in a dose-dependent manner from 52.5% with no EVs to 40% with 1× equivalent EVs and to 35.8% with 10× equivalent EVs (Figure 3e). Apoptosis is represented by morphological changes: cells were observed to turn phase-bright, followed by blebbing. Representative images show fewer phase-bright cells in the presence of EVs following treatment with gemcitabine and TRAIL (Figure 3f). 

### 2.4. Small EVs Decrease the Response of PDAC Cells to Gemcitabine and TRAIL Treatment 

As previously indicated, the EV fraction of CM contains several populations, including small EV particles approximately 50–150 nm in diameter [12]. We were interested in further investigating if these smaller populations were specifically responsible for the observed decrease in the response of PDAC cells to gemcitabine and TRAIL treatment, as so far described. We characterised the number of EVs using nanoparticle tracking analysis. From this analysis, it was shown that gemcitabine and TRAIL treatment increased the total number of EVs produced by MIA PaCa-2 cells by approximately 1.4-fold, while the neutral sphingomyelinase inhibitor GW4869 alone decreased the total number of EVs by approximately half (Figure 4a). GW4869 did not reverse the increase in total EVs following treatment with gemcitabine and TRAIL (Figure 4a). However, when we looked specifically at smaller EVs, the 1.7-fold increase in the number of particles following treatment was reversed following the addition of GW4869. The reduction in the total number of particles of this size is approximately 60% of that seen in control (Figure 4b). 

It is worth noting that GW4869 is known to be an exosome-specific inhibitor and was shown here to decrease the number of small EVs (Figure 4b). The remaining population of EVs, derived from the GW4869-treated MIA PaCa-2 cells, were then added with TRAIL onto PANC-1 cells. No significant difference in the percentage of cell viability was observed following this treatment (Figure 4c). Whilst we have so far considered the effects of incubating PDAC cells with CM or EVs derived from PDAC cells with differing responses to treatment, we also considered the effect of inhibiting the release of exosomes during treatment. This was investigated using MTT assays and by utilising GW4869 in combination with gemcitabine and TRAIL. A significant decrease in cell survival following treatment in the presence of GW4869 was noted in all three PDAC cell lines: BxPC-3, MIA PaCa-2 and PANC-1 (Figure 4d–f, respectively). 

## 3. Discussion

Both the late detection of PDAC and the occurrence of resistance to therapy have made PC a challenging tumour type. Resistance to gemcitabine has been well documented, both clinically and in pre-clinical models, and there has been an emergence of other therapeutics in the last decade [17]. The need for newly emerging treatments has formed the basis of our previous investigations, establishing that the use of the cancer-targeting cytokine TRAIL works synergistically with gemcitabine to increase cell death with treatment. Variability in the level of response between cell lines has been presented here; for example, PANC-1 cells exhibit much less cell death following treatment with the combination of gemcitabine and TRAIL compared with MIA PaCa-2. The implications of such observations in vitro have prompted us to explore these differences further. While not used here, technology such as self-sustaining nanoplatforms [18] could allow for the expansion of this work by reducing the overall resistance to TRAIL in cells. 

We were keen to characterise the role of CM and CM-derived EVs in the response of PDAC cells to gemcitabine and TRAIL treatment. In exploring the latter, it is important to note that EVs were also present in the serum used to supplement the growth medium. These have been shown to alter cellular characteristics such as proliferation and viability (reviewed extensively by Lehrich, Liang and Fiandaca [19]). Thus, comparisons were initially carried out to examine the effect of removing serum-derived EVs from the foetal calf serum (FCS) used to supplement the growth medium. While a significant decrease in cell survival was seen in response to the gemcitabine and TRAIL treatment in both MIA PaCa-2 and PANC-1 cell lines (Figure 1), it was found that the removal of serum-derived EVs did not alter the response (Appendix A). To test this further, we purified serum-derived EVs and included them in the treatment. This addition of ‘extra’ serum-derived EVs, however, altered the sensitivity of MIA PaCa-2 and PANC-1 cells to the combination treatment with gemcitabine and TRAIL, as seen by an increase in cell death as determined by the MTT assay (Appendix A). Therefore, all experiments were performed using media supplemented with EV-depleted serum. 

There have been several studies considering the effects of PDAC-cell-derived CM on other stromal components such as fibroblasts [11] and vice versa [20,21]. This work, however, is one of the first to consider the effects of PDAC-cell-derived CM on PC cells directly and specifically during gemcitabine and TRAIL treatment. Previous research has indicated that CM and PDAC-derived EVs taken from cells treated with gemcitabine are able to induce chemoresistance compared with those from untreated cells [22]. Leading on from this, our study has established the effects of both CM and PDAC-derived EVs on the response following combination treatment with gemcitabine and TRAIL. In particular, we characterised the effects of CM and PDAC-derived EVs from cells with differing responses on the sensitisation of PDAC cells during gemcitabine and TRAIL treatment. Unsurprisingly, CM derived from the cells that were more resistant to treatment, namely PANC-1, significantly increased cell viability in MIA PaCa-2 cells. Interestingly, the CM derived from the cells that were more sensitive to treatment, i.e., MIA PaCa-2, was also able increase cell viability in the more resistant PANC-1 cells. It should be noted, however, that this effect was not significant. Reassuringly, this latter observation indicates that the effects we have observed with the incubation of CM during TRAIL treatment are unlikely to be a result of nutrient depletion, given the short length of the treatment time (4–6 h) and given that generally it is accepted that less than 24 h should not affect viability excessively.

In looking more closely at the effects of CM on the response to treatment, we investigated the effects of the EV component of CM. Previous work has suggested that EVs are able to induce a change in chemosensitivity in the recipient cells by the delivery of molecules such as miRNAs and tumour suppressors [15,16]. In addition to the observed effects of PDAC-derived CM on cell viability following treatment with gemcitabine and TRAIL, we showed that CM-derived EV components can also inhibit the response of cells to treatment. EVs were characterised using nanoparticle tracking analysis (Figure 4a,b, discussed below), using a representative Western blot for the EV marker CD81 (Appendix A) and transmission electron microscopy (Appendix A). This was demonstrated by adding EVs derived from the more resistant cell line, PANC-1, to MIA PaCa-2 cells during treatment. Here, we saw a significantly reduced response, as determined by the decreased cleavage of caspase-8 (reduced apoptosis seen in comparison with that observed in the absence of EVs). Additionally, this was also reflected in an increase in MIA PaCa-2 cell viability following treatment with gemcitabine and TRAIL in the presence of PANC-1-derived EVs. The inhibitory effects of EVs on this response was further supported by similar observations using MIA PaCa-2 CM-derived EVs. Importantly, the effects characterised here were dose-dependent, as observed by a decrease in the percentage of apoptosis of PANC-1 cells. Such dose-dependent effects of EVs have also been previously shown in bladder cancer via the expression of apoptosis markers [23].

A sub-population of EVs was isolated by treating MIA PaCa-2 cells with the specific exosome inhibitor GW4869 [24]. This treatment resulted in a decreased proportion of EVs between 50 and 150 nm in diameter compared with the control untreated cells. Using the resulting sub-population of EVs, we have shown that when these were added to the TRAIL treatment of PANC-1 cells, there was no significant effect on cell viability. This contrasts with the effects observed when treating cells in the presence of the entire EV population. Not only does this indicate that the small-sized EV population is responsible for the effects but is also in agreement with other literature [16,25,26]. These studies have shown that the addition of small EVs to PDAC induces resistance in the recipient cells. Furthermore, we have shown that the inhibition of small EV release in all three PDAC cell lines during gemcitabine and TRAIL treatment significantly decreased cell survival. This suggests that the small EV population is likely to be responsible for mediating the response of cells to the treatment. To our knowledge, this has not been previously shown in PDAC but is in keeping with other literature that shows sensitisation to chemotherapies in other cancers, for example, to doxorubicin in acute myeloid leukaemia [27]. 

Interestingly, treatment of MIA PaCa-2 cells with the combination of gemcitabine and TRAIL significantly increased the number of small EVs. This is a novel finding that, coupled with the effects on the response, indicate that the EV population specifically plays a role in the resistance of PDAC cells to gemcitabine and TRAIL treatment. Several other studies have also identified that treatment with other potential chemotherapy agents (paclitaxel) [28] and proteosome inhibitors [29] increases the quantity of EVs released by tumour cells. Previous evidence, reviewed by Catalano and O’Driscoll, suggested that treatment with GW4869 increases the numbers of larger vesicles [24]. This effect was seen here when GW4869 was combined with treatment with gemcitabine and TRAIL (Figure 4a,b). Although beyond the scope of this study, the nature of the EVs released during the gemcitabine and TRAIL treatment of PDAC cells requires further investigation. 

One limitation of the current research is its focus on a few cell lines, which we have demonstrated to have varying levels of response to the combination treatment. To consolidate the findings shown here, further research should expand the number of cell lines used, in order to ascertain if these effects are consistent across all PDAC cells. However, despite these limitations, we present clear evidence that CM derived from untreated PDAC cells can confer resistance to treatment with gemcitabine and TRAIL. We show that this effect is due to the small EVs present in the CM. This provides a good foundation to further investigate the role of small EVs in the response of PDAC cells to treatment with gemcitabine and TRAIL. 

## 4. Materials and Methods

### 4.1. Materials

Unless specified, all reagents were purchased from Merck, Darmstadt, Germany. The biotinylated protein ladder; anti-caspase 8 (1C12) primary antibody; anti-rabbit, anti-mouse and anti-biotin immunoglobulin horseradish peroxidase secondary antibodies; and LumiGLO enhanced chemiluminescence kit were purchased from Cell Signalling Technology, Danvers, MA, USA. A full-range rainbow molecular weight marker, PVDF membranes and Hyperfilm were purchased from GE Healthcare, Chicago, IL, USA. 

### 4.2. Cell Maintenance

The PC cell lines BxPC-3, MIA PaCa-2 and PANC-1 were purchased from the ATCC. MIA PaCa-2 and PANC-1 cells were maintained in Dulbecco’s Modified Eagle Medium (DMEM) supplemented with 10–20% FCS, and penicillin and streptomycin, both at 50 units/mL. BxPC-3 cells were grown in RPMI-1640 medium supplemented with 20% FCS and 50 units/mL penicillin and streptomycin. Cells were maintained as a monolayer in humidified air at 37 °C with 5% CO_2_. Cells were treated with gemcitabine hydrochloride at a final concentration of 100 µM for a total period of 24 h, and with recombinant TRAIL (PeproTech EC, London, UK), which was added for 4 or 6 h at a final concentration of 100 ng/mL. 

### 4.3. Media Preparation

Prior to use on cells, the serum used to supplement DMEM was depleted of EVs utilising differential ultracentrifugation [30]. Briefly, FCS was diluted 1:1 with DMEM and then centrifuged at 300× *g* for 10 min, then 2000× *g* for 10 min, then 10,000× *g* for 30 min and finally 100,000× *g* for 70 min, maintained at 4°C; any pellets were discarded at each step. The final pellet generated was the EV-containing fraction. The FCS was filter-sterilised using a 0.2-micron filter and further diluted to the appropriate percentage in DMEM. The EV pellet was re-suspended in PBS and stored at −70 °C. 

To generate a cell-conditioned medium, MIA PaCa-2 and PANC-1 cells were seeded at a density of 3 × 10^4^ cells/cm^2^ and allowed to attach overnight. The DMEM was replaced with DMEM supplemented with EV-depleted FCS, and the cells were grown for 24 h. The cell-conditioned growth medium was then filter-sterilised using a 0.2-micron filter prior to addition to cells.

### 4.4. EV Isolation

EVs were isolated from the cell-conditioned growth medium using the process of differential ultracentrifugation [30] described above. The EV pellet was re-suspended in PBS and stored at −70 °C. EVs were added to cells relative to the volume of the cell-conditioned growth medium they were derived from (at 1× and 10×).

### 4.5. Tetrazolium Reduction (MTT) Assay

Cells were seeded at a density of 3 × 10^4^ cells/cm^2^ in 96-well plates and treated as required in quadruplicate. Wells were incubated with 25 µL thiazolyl blue tetrazolium bromide (MTT) reagent for 2 h, and any formazan crystals formed in this period were solubilised with SDS detergent overnight. Absorbance at a wavelength of 595 nm was recorded per well using a spectrophotometer (BioTek Synergy, Winooski, VT, USA, LX multimode reader) to allow quantitative determination of cell viability.

### 4.6. Trypan Blue Exclusion Assay

Cells were seeded at a density of 3 × 10^4^ cells/cm^2^ in 12-well or 24-well plates, and treated as required in triplicate. All media and cells were transferred to 1.5 mL tubes. A portion of this sample was combined in a 1:1 ratio with 0.4% Trypan Blue solution and vortexed to mix. Several counts were taken from each sample, and the percentage of viability was determined by the following formula: (number of white (viable) cells/total (white and blue) number of cells) × 100.

### 4.7. Time-Lapse Microscopy

The rate of cells committing to apoptosis was measured using time-lapse microscopy as previously described [31]. Briefly, cells were seeded at 3 × 10^4^ cells/cm^2^ in 12-well plates and treated as required in triplicate. Following addition of the treatment, cells were observed over a period of 24 h by an inverted Olympus Ix70 microscope enclosed in a chamber, where conditions were maintained at 37 °C and 5% CO_2_. Images were captured every 15 min of one field of view per well (randomly chosen at the start), containing around 40 cells. Analysis was carried out using ImageJ software to identify cells undergoing distinct morphological changes (shrinkage and turning phase-bright, followed by blebbing) as a marker for apoptosis. 

### 4.8. Western Blotting

Cells were seeded at 3 × 10^4^ cells/cm^2^ in dishes and treated as required. Cells were harvested and subject to lysis using a whole-cell lysis buffer supplemented with a protease inhibitor cocktail by vortexing on ice, followed by sonication. Samples were centrifuged at 14,000× *g* for 10 min at 4 °C. EVs were lysed using RIPA buffer supplemented with 1% Triton X-100 and 0.1% SDS. Samples were centrifuged at 14,000× *g* for 10 min at 4 °C before undergoing five freeze–thaw cycles. Following quantification, equal amounts of whole-cell extracts were subjected to polyacrylamide gel electrophoresis. Proteins were transferred to a PVDF membrane using semi-dry transfer and immunoblotted as previously described [32]. 

### 4.9. Nanoparticle Tracking Analysis

EVs isolated by differential ultracentrifugation were quantitated using nanoparticle tracking analysis. Samples to be quantitated were diluted 1:2000 and loaded into the NanoSight NS300 apparatus (Malvern Panalytical, Malvern, UK). The programme was set up to record each sample three times for a period of 1 min each time, and an average was taken. 

### 4.10. Transmission Electron Microscopy

After thawing, the EVs were resuspended in 100 mM sodium cacodylate buffer (pH 7.4). A drop (~3–5 μL) of the EV suspension was placed onto a TEM mesh grid with carbon support film, and the grid was previously glow-discharged to make the surface of the carbon film hydrophilic. The sample was then left to air-dry for ~10 min, followed by fixing for 1 min at room temperature by placing the grid onto a drop of a fixative solution (2.5% glutaraldehyde in 100 mM sodium cacodylate buffer (pH 7.0)). The grid was washed by applying it to the surface of three drops of distilled water, then removing excess water using a filter paper after each drop. Finally, the EVs were stained for 1 min with 2% aqueous uranyl acetate (Sigma-Aldrich, St. Louis, MI, USA), removing excess stain with a filter paper and air-drying the grid. EVs were imaged using a JEOL JEM 1400 transmission electron microscope (JEOL, Tokyo, Japan) at an 80 kV accelerating voltage, using a magnification of 30,000× to 60,000×. Recording of digital images was performed with an AMT XR60 CCD camera (Deben, Suffolk, UK).

### 4.11. Statistical Analysis

Data are presented as means ± SD. Any analysis undertaken was carried out using GraphPad Prism 9, including the use of one-way ANOVA (with post-hoc Dunnett’s tests) and Student’s *t*-test. Statistical significance was declared at *p* < 0.05.

## Figures and Tables

**Figure 1 ijms-23-07810-f001:**
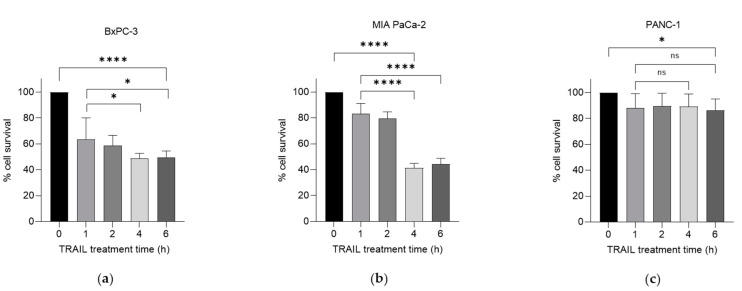
Effect of gemcitabine and TRAIL treatment on PDAC cell survival. The sensitivity of BxPC-3, MIA PaCa-2 and PANC-1 cells to the combination treatment was assessed using MTT assays. (**a**) BxPC-3, (**b**) MIA PaCa-2 and (**c**) PANC-1 cells were treated with 100 µM gemcitabine for 24 h and 100 ng/mL TRAIL for the final 0, 1, 2, 4, and 6 h of the gemcitabine treatment (*n* = 4). All experiments were repeated three times, and data are provided as means ± SD. One-way ANOVA followed by a *post-hoc* Dunnett’s test was used to determine statistical significance between gemcitabine alone or gemcitabine with 1 h of TRAIL and longer time periods with TRAIL: * *p* < 0.05, **** *p* ≤ 0.0001.

**Figure 2 ijms-23-07810-f002:**
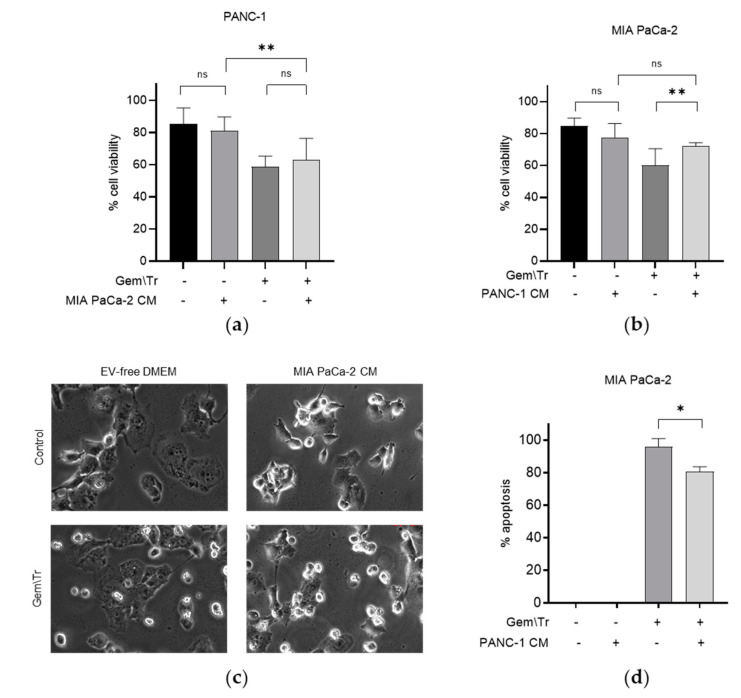
Effect of conditioned media on gemcitabine and TRAIL treatment of PDAC cells. The response of PDAC cells to treatment with 100 µM gemcitabine in combination with 100 ng/mL TRAIL over 4 h for MIA PaCa-2 cells and 6 h for PANC-1 cells is shown in Panels (**a**–**d**). Experiments were performed in the presence and absence of CM derived from cells showing differing responses, namely MIA PaCa-2 and PANC-1, and the effects on cell viability, morphology and apoptosis are shown. The sensitivity of cells to the combination treatment was assessed using trypan blue exclusion assays to determine the cell viability of (**a**) PANC-1 cells during treatment in the presence of MIA PaCa-2-derived CM and (**b**) MIA PaCa-2 cells during treatment in the presence of PANC-1-derived CM. (**c**) Cell morphology of PANC-1 cells in the presence of MIA PaCa-2-derived CM during treatment. (**d**) Time-lapse microscopy was used to determine the percentage of apoptosis in MIA PaCa-2 cells during treatment in the presence of PANC-1-derived CM. Data are displayed as means ± SD. For (**a,c**), one-way ANOVA followed by a post-hoc Dunnett’s test was used to determine statistical significance; for (**b**), Student’s *t*-test was used: * *p* < 0.05, ** *p* < 0.01. Gem\Tr, 100 µM gemcitabine with 6 or 4 h 100 ng/mL TRAIL; CM, conditioned media; EV-free, extracellular vesicle-free.

**Figure 3 ijms-23-07810-f003:**
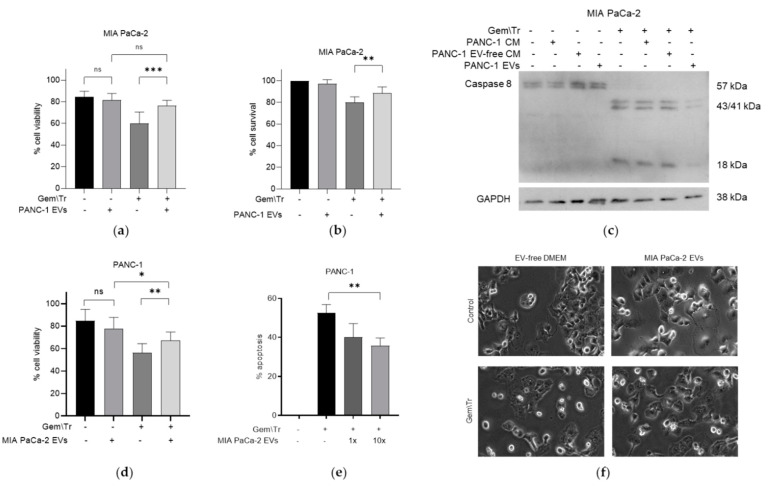
PDAC-derived EVs decrease the response of other PC cells to treatment with gemcitabine and TRAIL. PANC-1-derived EVs decrease the response of MIA PaCa-2 cells to gemcitabine and TRAIL, as assessed using (**a**) trypan blue exclusion assays to determine cell viability, (**b**) MTT assays to determine cell survival, and (**c**) Western blotting to identify caspase-8 cleavage. MIA PaCa-2-derived EVs decreased the response of PANC-1 cells to the combination treatment, as assessed using (**d**) trypan blue exclusion assays to determine cell viability, and time-lapse microscopy to determine (**e**) the percentage of apoptosis and (**f**) morphological changes. Data are displayed as means ± SD. One-way ANOVA followed by a post-hoc Dunnett’s test was used to determine statistical significance: * *p* < 0.05, ** *p* < 0.01, *** *p* < 0.001. Gem\Tr, 100 µM gemcitabine with 4 or 6 h of 100 ng/mL TRAIL; CM, conditioned medium; EVs, extracellular vesicles; EV-free, extracellular vesicle-free.

**Figure 4 ijms-23-07810-f004:**
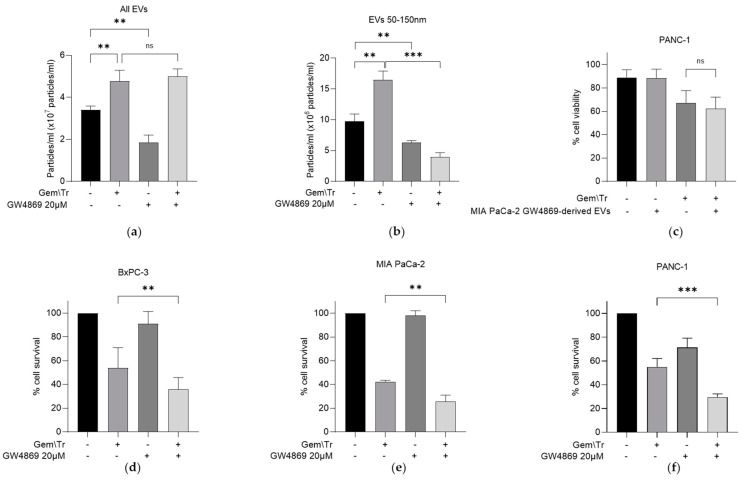
Effect of the sphingomyelinase inhibitor GW4869 on PDAC-derived EV production and on the cellular response of PDAC cells to gemcitabine and TRAIL. (**a**) MIA PaCa-2-derived EV production was analysed using nanoparticle tracking analysis to assess the effects of the sphingomyelinase inhibitor GW4869 (20 µM) during combination treatment with 100 µM gemcitabine and 100 ng/mL TRAIL. (**b**) Nanoparticle tracking analysis was repeated as described above to investigate the effects of the inhibitor on the production of smaller EVs derived from MIA PaCa-2 cells during combination treatment. (**c**) The effect of EVs derived from GW4869-treated MIA PaCa-2 cells on PANC-1 cell viability was investigated using trypan blue exclusion assays. PANC-1 cells were seeded as previously described and treated with the same combination of gemcitabine and TRAIL as described in (**a**). (**d**) BxPC-3 (**e**) MIA PaCa-2 and (**f**) PANC-1 cells were seeded as described above and treated with gemcitabine and TRAIL in the presence and absence of GW4869 as previously described. MTT assays were used to determine cell survival. Data are displayed as means ± SD. One-way ANOVA followed by a post-hoc Dunnett’s test was used to determine statistical significance in (**a**,**b**); for all others, Student’s *t*-test was utilised: ** *p* < 0.01, *** *p* < 0.001. Gem\Tr, 100 µM gemcitabine with 4 or 6 h 100 ng/mL TRAIL; EVs, extracellular vesicles.

## Data Availability

Not applicable.

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
