# Peer review of "Extracellular Vesicles Inhibit the Response of Pancreatic Ductal Adenocarcinoma Cells to Gemcitabine and TRAIL Treatment"

_ijms, 2022, doi:10.3390/ijms23147810_

Round 1

Reviewer 1 Report

The study of cell lines expanded on the combination of TRAIL and gemcitabine for pancreas cancer. The investigators could have examined gain of effect from enhancing TRAIL [1], but the study showed an inhibition and a derepressor of the inhibition. In short, Figure 4 indicate a synergy between GWA4869 and GemTr. But that is not underlined in abstract and text. Text is long and so is text for legends. The legends should better explain the content of the abbreviation used in the figures. Legends for Figure 2 to 4 should hint to abbreviations are explained in legend for Figure 1. Figure 1 shows discordance between viability an apoptosis. Legend for Figure 3 states that … ,and PANC-1-derived EVs decrease MAI PaCa-2 response to…. But the Figure shows cell survival not reduced survival. Figure 3e shows apoptosis where you had expected survival, and Figure 3e did not include the first of four columns. Figure 4 shows heterogeneity between all EVs and five cell lines. The report describes the observed effects of the experiments but could add thoughts of the biologic mechanisms leading to the observed laboratory findings for enhancing gemcitabine with TRAIL and GW4869. bibliography [1] Huang X, Ou C, Shu Y, Wang Y, Gong S, Luo R, et al. A self-sustained nanoplatform reverses TRAIL-resistance of pancreatic cancer through coactivating of exogenous and endogenous apoptotic pathway. Biomaterials 2021;272:120795.

Author Response

Reviewer 1 correctly states that this study has expanded on our previous research focusing on the combination treatment of TRAIL and gemcitabine in reducing cell survival and viability in PDAC. Whilst we could have chosen to reduce/ halt the effects described in this study, by altering the TRAIL response, this research focused on understanding how EVs play a role in the inhibition originally investigated.

Reviewer1 suggests the use of technologies such as the self-sustaining nanoplatform and we acknowledge that this would be an interesting area to explore. Although it is beyond the scope of this study, the reference they have suggested has now been incorporated and considered as part of our discussion section (lines 235-237).

With reference to Figure 4, Panels d-f show the effect of GW4869 and the combination treatment on cell survival as determined by MTT assays. Reviewer 1 suggests that there may be potential synergy between GW4869 and the combination of gemcitabine and TRAIL. We are not however able to strictly draw this conclusion based on those assays alone and would need to complete appropriate dose response curves, subjecting the data to Compusyn analysis. We have previously shown synergism between gemcitabine and TRAIL as a potentially new therapeutic. However, we did not aim to explore novel therapeutics in this manuscript and as such we feel that the conclusions drawn in our work and the extent to which we investigated the effect of GW4869 addition to the existing combination was sufficient. 

A list of abbreviations required to understand each figure has been included in the figure legend as required. These are provided as per the journals formatting guidelines and are specific to each figure. As such they vary according to the figure. We have taken on board the reviewers’ comments regarding Figures 2 and 4, expanding this to Figure 3, so that all figure legends now have appropriate abbreviations.

The reviewer suggests a discordance between viability and apoptosis in Figure 1. This figure however makes no reference to apoptosis. It refers only to cell survival as determined by MTT assay. It is Only Figures 2 and 3 that refer to apoptosis, where we used time lapse microscopy and western blotting to investigate this.

The legend for Figure 3 correctly states that “PANC-1-derived EVs decrease MIA PaCa-2 response to…”  This is shown by an increase in cell survival as determined by MTT assay (Figure 3b). In Figure 3e, the addition of EVs results in a decrease in the percentage of apoptosis, equating to the increase seen in cell survival. A small addition on line 150 in the results section clarifies these observations.

The data in this study was obtained using 3 PDAC cells lines. The reviewer indicates that Figure 4 refers to 5 cell lines. This is not the case, there are only 3 PDAC cell lines in this figure as a whole. The reviewer correctly states that there is some variability between the response of the different cell lines. We do not however see that this variability is significantly different enough to warrant further investigation. Indeed, we know from our previous studies using these cell lines that there is a difference in their response to gemcitabine and TRAIL treatment and in fact we have exploited this observation here to consider the effect of EVs on cells with differing response to that treatment.

Of note we have shown that EVs derived from PDAC cells exhibiting a varying sensitivity to gemcitabine and TRAIL treatment, all conferred the same ability- to inhibit the response to this treatment. A discussion on identification of the molecular mechanism underlying these effects has been included (line 273-274).

Reviewer 2 Report

In this paper, the authors have compared cell conditioned media from pancreatic cancer cells to “EVs” isolated by differential centrifugation. They are missing MISEV information, and have only shown the results, without clearly showing that it is EVs that is causing this protective effect on cells treated with Gem/TRAIL.

I think that with some modifications this can be published.

Overall it is well written, clear explanation of the experiments undertaken, and references previous work. However, it is quite a small study and its claims are missing some additional data.

This paper does not show any real evidence for EVs. Where is the electron microscopy of the EV pellet, or western blotting of the EV preparation?

Do you have Nanosight evidence for the GW4869 inhibition of EVs particularly on EVs 50 and 150nm in diameter? This paper by JEV appear to show that GW4869 may upregulate larger EVs reciprocally (Catalano, Mariadelva, and Lorraine O’Driscoll. "Inhibiting extracellular vesicles formation and release: a review of EV inhibitors." Journal of extracellular vesicles 9.1 (2020): 1703244.)

Is there any effect when Cells are treated with their own EVs? i.e. PANC-1 EVs with PANC-1 cells?

Do we know if this is a cancer cell specific effect?

Did you try other fractions of the differential centrifugation e.g. after a simple 10,000g spin?

Finally, I think “widely used first-line chemotherapeutic” is not the right term for gemcitabine, as it is now well-recognised as second line to FOLFURINOX after this trial. https://pubmed.ncbi.nlm.nih.gov/30575490/

Author Response

We have taken on board reviewer 2’s comments and now included Western blotting of the EV marker CD81 for the fraction isolated from MIA PaCa-2 cells in Supplementary figure 2. This, in addition to the Nanoparticle Tracking Analysis completed using the Nanosight strengthens the data showing the role of EVs in altering the response to gemcitabine and TRAIL treatment. It also brings our work in line with MISEV2018 guidelines. We have additionally referred to this in the discussion (Line 277-279) and also updated the materials and methods to reflect this inclusion (lines 387-389). 

Nanosight evidence for GW4869 inhibition of production of EVs between 50 and 150nm in diameter can be found in Figure 4b. The review by Mariadelva Catalano and Lorraine O’Driscoll is a very interesting read, and indeed when our PDAC cells were treated with both GW4869 and the combination treatment, we also observed a reciprocal increase in the larger sized EVs (as seen in Figure 4a and b, comparing the triple treatment for each population). This observation has now been highlighted in our discussion (lines 311-314).

At this stage we have not extensively considered the effect of treating PDAC cells with their own EVs (derived from untreated cells), as we were particularly interested in focusing the study on the effect of EVs derived from cells differing in response. We did, however, consider the effects of “self on self” using EVs derived from cells treated with the combination. In this case, those EVs had no significant overall effect on cell survival following treatment with the combination of gemcitabine and TRAIL. As such we felt it did not add to or detract from this narrative.  

This work focuses solely on the effect of PDAC-derived EVs on PDAC cells treated with gemcitabine and TRAIL. TRAIL can only target tumour cells through TRAIL surface receptors and as such has little effect on non-tumorigenic cells. Although we cannot say that the effects of EVs on response to treatment are only specific to cancer cells, we predict that EVs derived from PDAC cells would not have an effect on non-tumorigenic cells following similar treatment. This is largely due to a lack of response following TRAIL treatment in the latter cell type.  

We have not yet tried using other fractions of the differential centrifugation to investigate effects during TRAIL and gemcitabine treatment of PDAC cells. Using optimisation of a method by Thery et a.l, 2006), we elected to focus our study on EVs derived using this method. It would be interesting however to consider if other fractions have similar effects and something we can explore in future studies.   

Round 2

Reviewer 1 Report

I appreciate the will to add changes to the first draft.

Figure 3 still have 3 sets with 4 columns and 1 set with 3 columns.

There is no paragraph on limitations.

Conclusion is unchanged and weak.

If the study aimed to kill all cancer cells in the experimental setting, still the drugs used in the investigation allowed some cancer cells to survive.

Author Response

Figure 3 still have 3 sets with 4 columns and 1 set with 3 columns. 

Figure 3 a, b and d are data collected from trypan blue and MTT assays. The first column in each of these graphs shows the control untreated cells where survival and viability were largely unaffected. 

Figure 3e however is % apoptosis as determined from time lapse microscopy. The untreated cells did not demonstrate any apoptosis and so there is no column for that data. Cells that were not treated with gemcitabine and TRAIL remained healthy during the 24 h monitored. 

There is no paragraph on limitations.

While we had referred to some limitations throughout the work, we have now clearly addressed some limitations of the work – in particular the use of only a few cell lines – between lines 318 and 321. 

Conclusion is unchanged and weak.

We have strengthened the conclusions drawn, building off of the now included limitations, from lines 321-326.

If the study aimed to kill all cancer cells in the experimental setting, still the drugs used in the investigation allowed some cancer cells to survive.

TRAIL can target tumour cells through the DR4 and 5 receptors that are primarily found on many cancer cells types. However, previous work (and Figure 1 of this study) have demonstrated even within the three cell lines used, there is variability in response to this drug. This study did not aim to kill all of the cells; a significant decrease in survival post-treatment was necessary to ascertain if the addition of CM or EVs altered response. When beginning this research, the literature was not clear-cut as to whether EVs would be protective against or additive to the effects of treatment (and indeed, no literature existed with the combination used here). As such, the experimental design allowed for the assessment of a two-tailed test.